# Sequential conformational transitions and α-helical supercoiling regulate a sensor histidine kinase

Oskar Berntsson[1], Ralph P. Diensthuber[2], Matthijs R. Panman [1], Alexander Björling[1], Emil Gustavsson [1], Maria Hoernke[1,3], Ashley J. Hughes [1], Léocadie Henry[1], Stephan Niebling[1], Heikki Takala[1,4,5], Janne A. Ihalainen[4], Gemma Newby [6], Silke Kerruth[7], Joachim Heberle[7], Marianne Liebi[8], Andreas Menzel [8], Robert Henning[9], Irina Kosheleva[9], Andreas Möglich[2,10] & Sebastian Westenhoff[1]

Sensor histidine kinases are central to sensing in bacteria and in plants. They usually contain sensor, linker, and kinase modules and the structure of many of these components is known. However, it is unclear how the kinase module is structurally regulated. Here, we use nano- to millisecond time-resolved X-ray scattering to visualize the solution structural changes that occur when the light-sensitive model histidine kinase YF1 is activated by blue light. We find that the coiled coil linker and the attached histidine kinase domains undergo a left handed rotation within microseconds. In a much slower second step, the kinase domains rearrange internally. This structural mechanism presents a template for signal transduction in sensor histidine kinases.

[1] University of Gothenburg, 40530 Gothenburg, Sweden. [2] Humboldt-Universität zu Berlin, Berlin 10115, Germany. [3] Albert-Ludwigs-Universität Freiburg, 79104 Freiburg i.Br., Germany. [4] University of Jyväskylä, 40014 Jyväskylä, Finland. [5] University of Helsinki, 00014 Helsinki, Finland. [6] European Synchrotron Radiation Facility, 38000 Grenoble, France. [7] Freie Universität Berlin, 14195 Berlin, Germany. [8] Paul ScherrerInstitut, 5232 Villigen, Switzerland. [9] The University of Chicago, Chicago, IL 60637, USA. [10] Universität Bayreuth, Bayreuth 95440, Germany. Oskar Berntsson and Ralph P. Diensthuber contributed equally to this work. Correspondence and requests for materials should be addressed to A.M. (email: andreas.moeglich@uni-bayreuth.de) or to S.W. (email: westenho@chem.gu.se)

Sensor histidine kinases (SHKs) constitute the main means by which bacteria gather information about their surroundings. They are part of so-called two-component systems and are also found in plants and certain other eukaryotes[1, 2]. SHKs detect external signals, for instance chemicals, light, or pH changes. This triggers structural changes and the activity of the histidine kinase output domains is modulated.

The architecture of SHKs is now well understood, and crystal structures of several SHK fragments have been elucidated[3]. Prototypical SHKs are membrane-spanning homodimeric proteins, consisting of an extracellular sensor module, a helical domain traversing the plasma membrane, an optional PAS, GAF, or HAMP domain, a coiled coil linker, and finally the "dimerization and histidine phosphotransfer" (DHp) and "catalytic and ATP-binding" (CA) domains, which together form the histidine kinase effector module[3]. Building on this foundation, recent focus has been on structurally characterizing the phosphoryl transfer[4, 5], on the interaction with phosphoacceptor proteins[6], and on how structural changes control the switching between phosphatase and kinase activity[7].

Another important question is to find out how external signals structurally modulate the activity of the histidine kinase domains. A number of conflicting mechanisms have been proposed. Using crystal structures and disulfide cross linking, transmembrane signaling has originally been ascribed to a swinging piston mechanism[8] and more recently to diagonal displacement in the transmembrane four helix bundle[9]. Further downstream, the transduction of signals into the histidine kinase module has been proposed to occur according to a gearbox model[10], through helix bending[11], partial unfolding of the DHp domain[12], a torque motion of the linker helices[13, 14], or through a combination of these models[4]. Only two of these proposals were made based on crystal structures of intact SHKs containing both sensor and effector domains[11, 13], but the remainder are based on the structures of isolated sensor, linker and kinase domains, or on biochemical data. There is a dearth of data reporting on the structural change between the activated and resting states, and the signal transduction mechanism remains contentious.

Exploiting the modular architecture of sensor histidine kinases and light-oxygen-voltage (LOV) photoreceptors[15], the chimeric SHK YF1 was generated via fusion of the *Bacillus subtilis* YtvA LOV photosensor domain and the histidine kinase module of FixL from *Bradyrhizobium japonicum*[16]. YF1 is a fully functional histidine kinase, and its biochemical activity is modulated by blue light. As a light-regulated actuator, it underpins several applications in optogenetics[17, 18]. In the dark YF1 is in a kinase-active state. Excitation of a flavin mononucleotide chromophore within the LOV domain by blue light creates first an excited singlet state (FMN*), which rapidly decays into a triplet state (FMN$^T$). The signaling state (FMN$^{A390}$) is then generated by formation of a thioether bond between the flavin and a conserved cysteine residue of the LOV protein. YF1 is thereby transformed into a phosphatase-active state. The homodimeric YF1 receptor serves as a soluble model of the intracellular part of the SHK family.

The crystal structure of dark-adapted YF1 shows how two LOV photosensor domains form a dimer, which is anchored together by two short N-terminal coiled helices, denoted A'α (Fig. 1a)[13]. A second coiled coil (Jα) is coaxial with A'α and links to the all-helical DHp domain of the histidine kinase module. Notably, the protein is kinked and has asymmetrically arranged CA domains. Both features are typical for histidine kinases in their kinase-active state, such as dark-adapted YF1, and likely have functional relevance[3, 11, 19, 20], but may also be crystallization artifacts. Two recent investigations by electron paramagnetic resonance spectroscopy[21] and X-ray solution scattering[22], respectively, consistently revealed that the LOV dimer in YF1

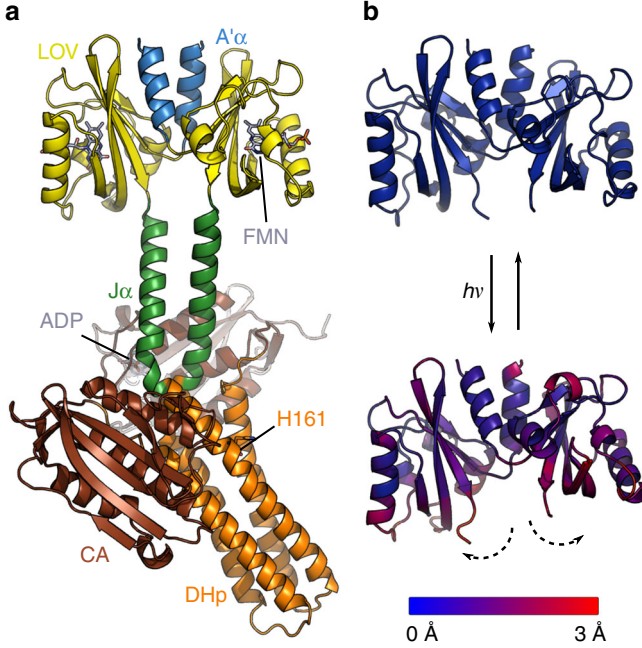

**Fig. 1** Structure of YF1. **a** Structure of YF1 in its dark-adapted state as resolved by X-ray crystallography[13]. The location of the different domains, of the flavin mononucleotide (FMN), of the cofactor adenosine diphosphate (ADP), and of the phosphoaccepting histidine 161 are indicated. **b** Light induced conformational changes of the LOV photosensor domain refined from X-ray solution scattering[22]. The changes are maximal at the C-termini that feed into the Jα helices (dashed arrows). The coloring is according to the root mean square deviation of the alpha carbons

pivots apart upon blue-light exposure (Fig. 1b). As a result, the C-termini of the LOV dimer, which are the connection points of the Jα helical linker, move apart by 3 Ångströms. Here we use time-resolved solution X-ray scattering[23] to investigate how these structural transitions propagate in full-length YF1. We identify the conformational changes and the sequence of events by which the kinase is regulated in solution.

## Results

**Time-resolved X-ray solution scattering.** Time-resolved X-ray solution scattering measures the change in distance between all scattering electrons of the sample. The data thus provide information on conformational changes of the sample. The method is particularly suited to interrogate structural changes of transiently populated states of proteins in solution[23–26]. We hence recorded time-resolved X-ray solution scattering of YF1 on nanosecond to second time scales after excitation with a blue laser pulse (Fig. 2). All experiments were carried out in the absence of adenosine nucleotides. Typically, time-resolved X-ray scattering data contain contributions from heated solvent. This effect was hardly seen in our data, but we still subtracted the contribution following standard procedure (Supplementary Fig. 1). Any concentration dependence of the scattering curves was below the $q$-range used in the analysis (Supplementary Fig. 2; $q$ is the modulus of the scattering vector).

Difference signals are observed for delay times >250 ns in the range 0.06 Å$^{-1}$ < $q$ < 1 Å$^{-1}$ (Fig. 2). Peaks in this $q$-range indicate that changes occur on all length scales in the protein. With delay time, the signal increases in amplitude and changes shape until it reaches a steady state at ~300 ms after laser excitation. We found that two sequentially populated states are significant and sufficient to describe the time evolution of the data. This is seen

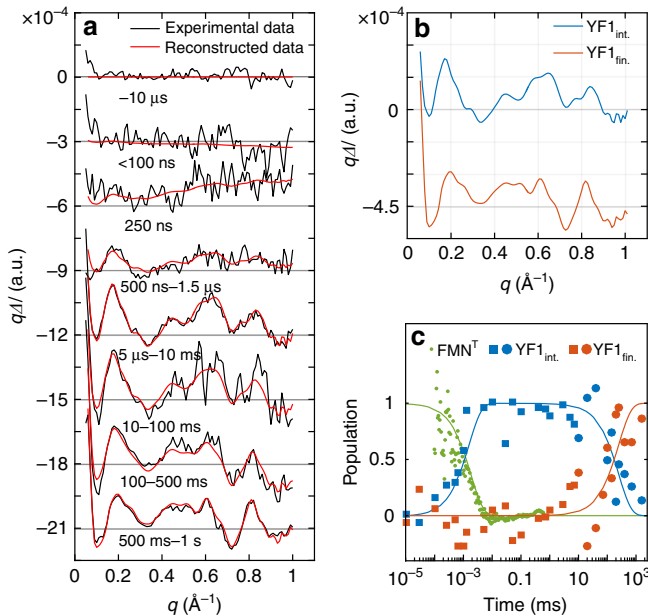

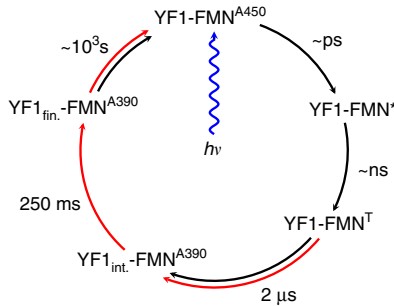

**Fig. 2** X-ray solution scattering data on YF1. **a** Experimental difference X-ray scattering ($\Delta I$), combined from two separate experiments, covering time delays from nanoseconds to milliseconds (recorded at BioCARS of the Advanced Photon Source) and from millisecond to seconds (recorded at cSAXS of the Swiss Light Source), respectively. The modulus of the scattering vector is defined as $q = 4\pi\sin(\theta)/\lambda$, with $2\theta$ as the scattering angle and $\lambda$ as the X-ray wavelength. Timepoints labeled e.g., 10–100 ms indicate that several delay times between the two time points are averaged together. The data was decomposed into a model which assumes two sequentially populated states, $YF1_{int.}$ and $YF1_{fin.}$ (see main text for details). The basis patterns are shown in **b** and the kinetics in **c** (lines). The reconstructed data in **a** were computed from a linear combination of the time-independent basis spectra, with the squares and circles in **c** representing the coefficients for the reconstruction of the data recorded at BioCARS and cSAXS, respectively. The green dots show the decay of the triplet state as measured by flash photolysis absorption spectroscopy at a wavelength of 715 nm. The line is a fit to an monoexpoential decay law.

**Fig. 3** Structural photocycle of YF1. Within pico- to nanoseconds after illumination, all rearrangements are confined to the chromophore and its immediate surroundings (YF1-FMN*, YF1-FMN$^T$). The flavin triplet state (YF1-FMN$^T$) decays into the thioadduct state, which is synchronous with protein conformational changes (YF1$_{int.}$-FMN$^{A390}$). The protein conformation changes again within milliseconds, without any response from the chromophore (YF1$_{fin.}$-FMN$^{A390}$). Eventually the resting state (YF1-FMN$^{A450}$) is recovered. Spectroscopic transitions of the chromophore are marked as *black arrows* and structural transitions of the protein as measured by X-ray solution scattering are marked as *red arrows*, respectively.

in Fig. 2c, where agreement is observed between the lines (model) and the dots, which represent the linear combination of the basis spectra (Fig. 2b). Further, the "reconstructed data" in Fig. 2a reproduce the data well. An intermediate state ($YF1_{int.}$) arose with a time constant of 2 μs and decayed into the final state ($YF1_{fin.}$) with a time constant of 250 ms (Fig. 2b, c). The transition from $YF1_{int.}$ to $YF1_{fin.}$ is characterized by a shift of a prominent peak along $q$ from 0.17 Å$^{-1}$ to 0.19 Å$^{-1}$ and by changes in the oscillatory pattern between 0.4 Å$^{-1}$ to 0.8 Å$^{-1}$. This is further illustrated by comparing the isolated difference curves at 50 μs and 3–4 s (average) shown in Supplementary Fig. 3. Although a linear two-step mechanism adequately describes the experimental data, a more complex kinetic model cannot be excluded. In particular, other experimental techniques may be able to resolve additional intermediates as has for example been the case for related LOV photoreceptors[27–30].

To correlate these structural events with photochemical changes of the chromophore, we measured the time-resolved absorption of YF1 after blue-light exposure (Fig. 2c, green data points and line). Kinetics at 715 nm reflect the decay of the triplet state FMN$^T$ and the formation of the thioadduct photoproduct (FMN$^{A390}$). The data reveal that this photochemical transition is synchronous with the formation of the intermediate state $YF1_{int.}$ as detected by X-ray scattering. In contrast, the formation of the

final structural state $YF1_{fin.}$ is not associated with any detectable changes in UV/vis absorption spectroscopy.

Based on these findings, we propose a structural photocycle of signal transduction in YF1 (Fig. 3). After absorption of a photon by the dark-adapted state (FMN$^{A450}$), the FMN chromophore and its immediate surroundings respond photochemically (FMN*, FMN$^T$)[31, 32]. Then difference X-ray scattering detects a first global conformational change yielding $YF1_{int.}$. The structural change occurs simultaneously with formation of the flavin-cystein thioadduct (FMN$^{A390}$) with a time constant of 2 μs. On the millisecond time scale YF1 undergoes additional conformational rearrangements to form $YF1_{fin.}$, which are not accompanied by further changes in the chromophore. We have recently reported that the isolated LOV photosensor fragment of YF1 does not have a corresponding late transition[22] and we therefore suspect that this structural change localizes to the linker and histidine kinase domains. Finally, the protein and the chromophore return to their respective resting states (YF1-FMN$^{A450}$).

**Structural analysis.** We employed reverse modeling of the time-resolved X-ray scattering data to extract the structural changes of $YF1_{int.}$ and $YF1_{fin.}$[26]. In short, we generated two trajectories representing the dark- and light-state structures of the protein using molecular dynamics (MD) simulations. We then computed the X-ray difference scattering between all generated "light" and "dark" structures, compared to the experimental data, and selected the best fits.

The MD trajectories were generated starting from the crystal structure of dark-adapted YF1 (pdb ID: 4GCZ)[13]. In order to overcome the limited sampling of MD simulations, we biased two trajectories towards the "dark" and "light" structure by restraining the alpha carbons (Cα) of the LOV domain to the respective position of the dark and light adapted LOV domain. These restraints were based on our previous studies on the LOV domain of YF1, which revealed that the two LOV monomers pivot apart in response to light[21, 22]. The difference scattering of all 2 billion combinations of dark and light structures (45,000 each) was computed and compared to the experimental data. The 100 pairs that best described the experimental data, as determined by lowest $R$ value (cf. Methods, Eq. (3)), were selected as good fits for $YF1_{int.}$ and $YF1_{fin.}$, respectively (Fig. 4a, b). From the laser energy titration (Supplementary Fig. 4) we concluded that the difference

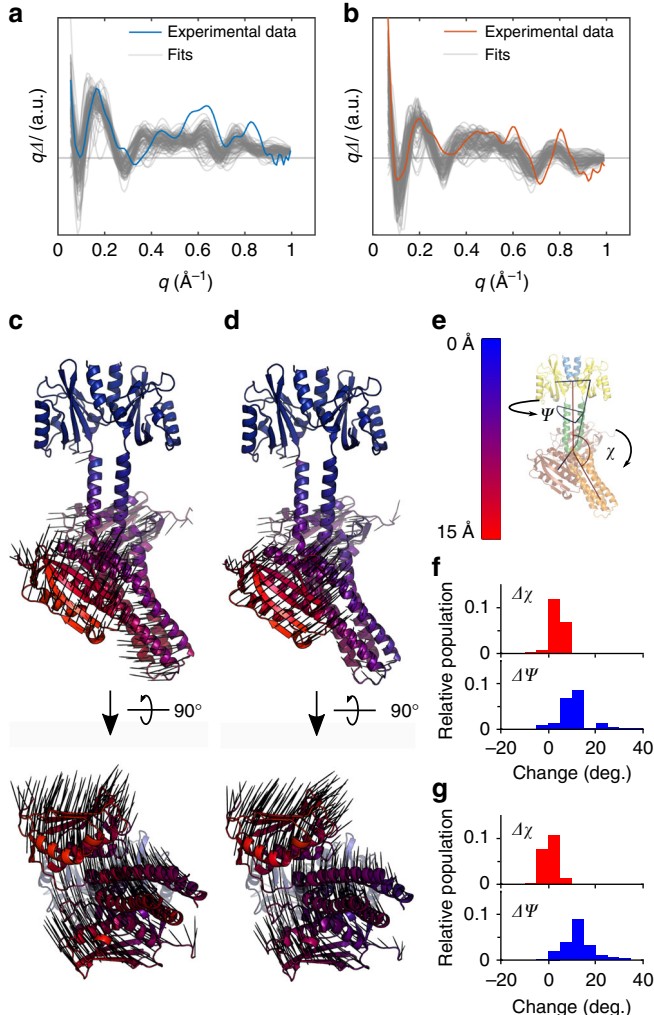

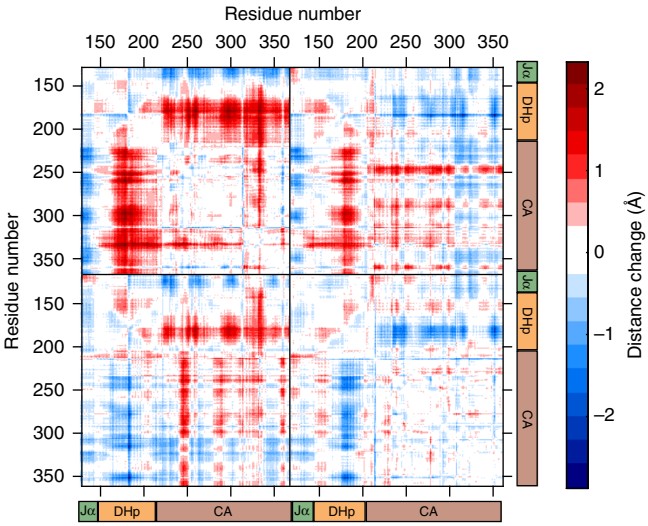

**Fig. 5** Cα double distance difference matrix. The difference in Cα distance changes between YF1$_{int.}$ and YF1$_{fin}$

**Fig. 4** Structural modeling of X-ray scattering data. **a** and **b** Theoretical difference scattering of the best-fitting pairs and experimental difference scattering for YF1$_{int.}$ (**a**) and YF1$_{fin.}$ (**b**). **c** and **d** Schematic representations showing the amplitude of motion relative to the dark-adapted state for YF1$_{int.}$ (**c**) and for YF1$_{fin.}$ (**d**). Movements within the LOV domain are a few Ångströms at most, which makes it appear almost rigid in the context of full-length YF1. **e** A cartoon depicting the angle (χ) and dihedral angle (ψ). The arrow tips indicate the direction of positive rotation around χ and ψ (see main text for more precise definition). **f** and **g** The distribution of changes in χ and ψ for YF1$_{int.}$ (**f**) and for YF1$_{fin.}$ (**g**)

curves represent on the order of 70% excitation efficiency. The modeled data reproduced the peaks very well at $q < 0.25$ Å$^{-1}$. However, at higher $q$ some deviations occur. Since $q$ is roughly inverse to distance, we expect the models to represent global structural change well, whereas some smaller scale local structural rearrangements may not be accurate in all structural models.

All pairs of structures were overlaid and visually inspected to identify a common structural change, but the resting and photoproduct state structures did not group into two distinguishable sets of structures. Considering that the data report on differences rather than on absolute structures this is not very surprising. Instead, we aligned all accepted pairs on the LOV domain and computed the mean displacement of the Cα atom of each residue. The structures in Fig. 4c, d are color coded according to this parameter. The arrows visualize the direction of this mean displacement. Taken together, they indicate a left handed rotation of the kinase effector module with respect to the sensory module in YF1$_{int.}$ and YF1$_{fin.}$.

To further quantify the structural change, we defined the angles χ and ψ, which report on the bending and torsion of the histidine kinase domains with respect to the LOV domains (Fig. 4e). The angles are defined as follows: χ is the angle between the center of mass of Cα of residue 11–127, 146–156, and 170–200 and ψ is the dihedral angle between the center of mass of Cα of residue 11–127 of monomer A, 11–127 of monomer B, 224–377 of monomer A, and 224–377 of monomer B. The average changes in χ and ψ between the selected sets of structures of the dark-adapted state and the state YF1$_{int.}$ indicate a very slight straightening of the entire protein by ca. 2° and a larger left-handed rotation of the kinase domain with respect to the LOV domain by 11° (Fig. 4f). Structures selected for YF1$_{fin.}$ showed a similar change in the bend of the protein and a somewhat more pronounced rotation of 12° (Fig. 4g). This analysis confirms that both states are rotated left-handedly with respect to the resting state.

YF1$_{int.}$ and YF1$_{fin.}$ show similar overall conformational rearrangements and to better assess the difference between the two, a Cα double distance difference matrix was computed for the kinase module. That is, the Cα distance difference matrix for YF1$_{int.}$ with respect to the dark state was subtracted from the Cα distance difference matrix for YF1$_{fin.}$with respect to the dark state (Fig. 5). The figure shows that one of the CA domains moves closer to the helices of the DHp domain (blue data points in Fig. 5) whilst the other moves away from them (red data points in Fig. 5). Thus, we conclude that an internal rearrangement of the kinase module occurs between YF1$_{int.}$ and YF1$_{fin.}$.

The present data show that (i) the rise of YF1$_{int.}$, but not of YF1$_{fin.}$, coincides with a change in the chromophore (Fig. 3) and a structural transition in the isolated LOV domain[22]; (ii) in both YF1$_{int.}$ and YF1$_{fin.}$the kinase module is rotated with respect to the sensory module (Fig. 4); and (iii) the histidine kinase domains rearrange internally between YF1$_{int.}$and YF1$_{fin.}$(Fig. 5). For these reasons we conclude that the formation of the signaling state in the LOV domain directly causes left-handed supercoiling of the alpha helical linker, thereby enabling the slower internal rearrangement of the histidine kinase domains (Fig. 6).

As a word of caution we note that the details of the refined structural change for YF1$_{fin.}$(Fig. 4) may depend on the choice of the resting state structure. We used the crystal structure of dark-adapted, kinase-active YF1 (pdb ID: 4GCZ), which, as described above, is kinked and possesses asymmetrically arranged CA

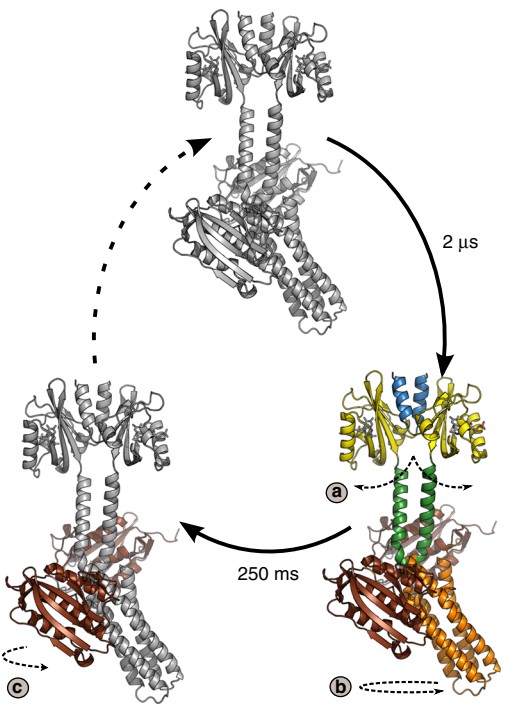

**Fig. 6** Structural photoactivation of YF1. Blue light causes sequential structural arrangement in YF1. After 2 μs the LOV domains splay apart (arrow a), causing supercoiling of the Jα helices and the rotation of the entire kinase module (arrow b). After 250 ms the kinase domain rearranges internally as the CA domains relocate on the DHp helices (arrow c)

domains[13] (cf. Fig. 1a). We note that the straightening in YF1$_{int.}$ may be affected by the kink in the YF1 structure, but that the detected rotation is likely not. The rotation manifests itself as signals at $q < 0.25$ Å$^{-1}$ and is clearly observable in the difference scattering because the electron density of the bulky CA domains moves. We do not expect that hypothetical straightening out of the protein would change this finding.

## Discussion

Our X-ray scattering data reveal that the photoinduced structural change of the LOV domain (arrow a in Fig. 6) leads to left-handed supercoiling of the coiled coil Jα helices in the linker (arrow b). Supercoiling may be accompanied by unbending of the DHp domain which is kinked in the dark-adapted YF1 structure. Although our present scattering data only provide tentative evidence for such a straightening (cf. Fig. 4f, g), it is well documented for other SHKs[3, 19]. These conformational rearrangements in YF1 occur concomitantly with formation of the chromophore photoproduct within microseconds of light absorption.

All scattering experiments were performed on full-length YF1 in the absence or presence of its cognate signal blue light. Based on the biochemical characterization of YF1[16], we can hence unequivocally assign the resting, dark-adapted structural state of YF1 as kinase-active and the long-lived structural state YF1$_{fin.}$ populated after blue-light exposure as phosphatase-active. Due to its transient nature, we cannot functionally assign the YF1$_{int.}$ state.

We propose that left-handed supercoiling is the structural mechanism by which signals are relayed from the sensor to the effector module of the sensor histidine kinase and by which SHK activity is eventually switched from kinase to phosphatase. Supercoiling is in agreement with a recent biochemical study, which found a twist in the related coiled coil HAMP domains in

the context of the histidine kinase PhoQ[14]. Crucially, left-handed supercoiling of the linker and connected DHp helices implies a rotation of the individual helices. Recent work on the DesK histidine kinase from *B. subtilis* indeed shows such concerted torque and rotary movements in the phosphatase-active state, cf. Fig. 4c of Trajtenberg *et al.*[7]. As a consequence of these movements, the active-site histidine is rotated away and thereby sequestered in the interior of the antiparallel four-helix bundle of the DHp domain, whereas residues important for catalyzing the phosphatase reaction are moved into proper position.

Light-induced supercoiling of the Jα coiled coil provides the structural rationale for a slew of recent functional data on YF1 variants, where insertion of single residues into the linker sufficed for inverting the response to light[33]. Within a coiled coil, insertion of one residue amounts to a change of angular orientation, which apparently controls histidine kinase activity. Left-handed supercoiling suggested by the present structural data promotes exactly such angular reorientation in the coiled coil linker. Our data also provide insight into the mechanism of functional variants of YF1 that bear single mutations within its LOV photosensor. The exchange of residue H22 at the junction between A'α and the LOV core domain to proline caused inverted signal polarity[13], in that the resultant YF1 variant H22P assumes its kinase-active state under blue light rather than in the dark. Electron paramagnetic resonance spectroscopy[21] showed that although the H22P variant has a drastically altered dimer interface it yet undergoes a light-induced pivot motion akin to that in YF1 albeit with inverted sign. In light of the present findings, we propose that the inverted signal polarity of H22P might be due to an inverted, i.e., right-handed, α-helical supercoiling upon light absorption.

With respect to YF1$_{int.}$ the scattering data reveal an internal repositioning of the CA domains within the histidine kinase module (arrow c in Fig. 6). Notably, only certain DHp/CA binding geometries are competent for phosphate transfer to the active-site histidine residue[3]. Reorientation of the active-site histidine and the CA domain has been implicated as a key mechanism for regulating phosphorylation[12]. The data presented here strongly support this proposal.

We propose that the light-induced change in supercoiling evidenced in YF1$_{int.}$ and YF1$_{fin.}$ alters the binding interface between the DHp and CA domains, thus promoting detachment and relocation of the latter. This view is supported by that the formation of the intermediate (YF1$_{int.}$) and final states (YF1$_{fin.}$) are hierarchically separated in space and time (Fig. 6): the first step is complete within microseconds of light absorption and is mainly confined to the chromophore, the LOV photosensors, and the linker. By contrast, the second step displays kinetics that are five orders of magnitude slower, and the conformational changes are internal to the histidine kinase moiety. We note that signal-dependent modulation of the DHp:CA interface represents a general and crucial feature observed for numerous SHKs[3] including an unusual monomeric LOV-coupled histidine kinase from a marine bacterium[34]. Our work now reveals how detection of external signals, here blue light, by the sensor module of SHKs can be structurally transduced to the SHK effector to remodel the DHp:CA interface.

Sequence data provide clear-cut evidence that the present-day plethora of signal receptors largely results from recombination of much smaller sets of sensor, linker, and effector modules[35]. Although YF1 is a chimeric receptor, its construction employed the same principle of modular recombination and its domain architecture corresponds to that of numerous natural receptors. We therefore suggest that the spatial and temporal hierarchy of the signal transduction trajectory in YF1 more generally reflects the modular architecture and evolutionary origin of signal

receptors. The universal presence of coiled coil linker domains in SHKs leads us to propose that supercoiling is a wide-spread means of transducing signals in this protein family.

## Methods

**Protein production and purification.** YF1 was produced in *Escherichia coli* BL21 CmpX13 cells and purified using affinity chromatography, as described in Diensthuber et al.[13]. In brief, *E. coli* BL21 CmpX13 cells[36] were transformed with the pET-41a-YF1 expression plasmid and were grown at 37 °C and 225 pm in Luria broth supplemented with 50 µg ml⁻¹ kanamycin until an optical density at 600 nm of 0.6 was reached. Induction was achieved by addition of 1 mM β-D-1-thioga-lactopyranoside, and cells were incubated for an additional 4 h at 37 °C and 225 pm. Cells were pelleted by centrifugation, resuspended in buffer A (50 mM Tris-HCl pH 8.0, 20 mM NaCl, 20 mM imidazole, protease inhibitor cocktail Complete Ultra (Roche Diagnostics)) and lysed by ultrasound. After clearing by centrifugation, the lysate was applied to a 5 ml Ni²⁺-chelate affinity column (GE Healthcare). Following extensive washing, the protein was eluted from the column by a gradient from 20 mM to 1 M imidazole over 60 ml. Elution fractions containing YF1 were identified by denaturing polyacrylamide gel electrophoresis and pooled. After dialysis into storage buffer (10 mM Tris-HCl pH 8.0, 10 mM NaCl, 10% (v/v) glycerol), the protein was concentrated by spin filtration. Protein concentration was determined spectrophotometrically using an extinction coefficient of 12,500 M⁻¹ cm⁻¹ at 450 nm. Upon flash freezing in liquid nitrogen, aliquots were stored at −80 °C.

**Time-resolved UV/vis absorption.** Time-resolved UV/vis absorption measurements were conducted as described in Kerruth et al.[37]. Briefly, a LKS80 Laser Flash Photolysis Spectrometer (Applied Photophysics) was used to excite a solution of 67 µM YF1 by a 10 ns laser pulse at 450 nm (Nd:YAG driven OPO, 4 mJ cm⁻²). Time-dependent absorption changes were probed with a pulsed Xe arc lamp and were recorded at 715 nm with a photo multiplier tube. Data from 10 individual measurements were averaged.

**X-ray scattering data acquisition.** X-ray solution scattering data were collected at the beamlines 14-ID-B/BioCARS at the advanced photon source (APS), at X12SA/cSAXS at the Swiss light source (SLS). Unless otherwise stated, typical protein concentrations were 20–23 mg ml⁻¹ in a buffer containing 10 mM Tris pH 8.0, 10 mM NaCl and 10% glycerol. Because the water scattering shows an isobestic point with respect to heating at $q \sim 1.5$ Å⁻¹, the scattering curves were normalized to the scattering in the range 1.4 Å⁻¹ < q < 1.6 Å⁻¹, according to common practice (e.g., refs [23, 24, 26]). Difference scattering curves representing pure heating of the protein solution were collected by exposing the protein solution to an infrared laser pulse (1470 nm), or a series of such pulses. The number and duration of pulses were adjusted to generate difference scattering in the region 1.5 Å⁻¹ < q < 2.2 Å⁻¹, where almost all signal is due to the solvent, with the same amplitude as in the blue-light excited difference data. These curves were then scaled to and subtracted from the blue-light excited difference scattering curves before further analysis.

Laser pump X-ray probe experiments using integrating CCD detectors were performed at BioCARS using an X-ray energy of 12 keV. The X-rays were focused to a ca. 60 × 90 µm spot. The sample was excited using a 450 nm, 5–6 ns laser pulse with an energy density of 0.5–1 mJ mm⁻², the laser spot size was adjusted so that it engulfed the X-ray spot. To collect data at different time points after excitation the temporal offset between laser and X-rays was adjusted. The X-rays and laser were overlapped on a 1 mm diameter quartz capillary, holding the sample. The protein was delivered through inert PEEK tubing using a syringe pump at a constant pump speed of 2–5 µl s⁻¹. When not in the tubing or capillary the sample was stored in a reservoir, kept at 30 °C to allow the sample to revert to the dark state. To exclude that the recorded signal was generated by the addition of heat to the sample, and to enable subtraction of the solvent contribution to the difference scattering, an experiment with infrared (1470 nm) excitation was performed. This experiment was executed in the same way as described, the difference being that the sample was excited with laser pulse at a 1470 nm and a duration of ~10 ms. Detector images were radially integrated using software at the beamline. When using a syringe pump images recorded close to the turn of the pump were corrupted. These were manually removed. Difference scattering data deviating from the median by more than 3σ (σ is the standard deviation of the different scattering curves) in the region 1.7 Å⁻¹ < q < 2 Å⁻¹ were rejected, this was done on a per delay time basis. Scattering curves corresponding to the same X-ray-laser delaytime were averaged together.

X-ray scattering experiments using rapid readout photon counting detectors were performed at cSAXS using an X-ray energy of 11.2 keV. The X-rays were focused to a ca. 100 × 200 µm spot. Excitation was achieved using a 457 nm, 10 ms long laser pulse, with an energy density of ca. 10 mJ mm⁻². X-rays and laser was overlapped on a 1 mm diameter capillary. The detector was read out at 25 Hz over ca. 5.5 s. The protein was delivered through PEEK tubing using a peristaltic pump. In contrast to the experiment at BioCARS the pump was stopped during data acquisition. When not in the tubing or the capillary the protein was kept at 30 °C. Heating of the sample was achieved not by a single pulse as at BioCARS but

through a train of short (≈µs) pulses over 2 s, yielding a stable addition of power. Detector images were radially integrated using software at the beamline. Data deviating from the median scattering in the by more than 5% in the region 1.7 Å⁻¹ < q < 2.2 Å⁻¹ were considered outliers and were removed before further analysis. Scattering curves corresponding to the same X-ray-laser delay time were averaged together. During this experiment a laser energy titration was also performed. This was done to investigate what proportion of the protein was excited by the laser and was achieved by tuning the length of the laser pulse. Difference scattering for a dilution series was performed to asses the effects of high protein concentrations. Protein concentrations of ca. 2.5, 5, 10 and 21 mg ml⁻¹ were measured. For these measurements all delaytimes were averaged together.

**X-ray scattering data analysis.** The time-resolved scattering data of recorded at BioCARS and at cSAXS were merged to allow analysis of the entire time range, from ca. 100 ns to ca. 5 s. Since the X-rays at BioCARS are polychromatic, whilst the X-rays at cSAXS are monochromatic the cSAXS curves were convoluted with the BioCARS undulator spectrum according to established methods[23, 24, 38–41]. As an illustration the same operation was performed on the steady state difference scattering for the isolated YF1-LOV domain[22] (Supplementary Fig. 5), showing that this procedure reproduces the polychromatic difference curve with high accuracy. The time dependent scattering data is considered to be a linear combination of time independent basis spectra according to Eq. (1).

$$q\Delta I(q,t) = \mathrm{BS}(q) \cdot C(t) \tag{1}$$

Where $\Delta I(q,t)$ is the experimentally recorded data, $\mathrm{BS}(q)$ is the time independent basis spectra and $C(t)$ is the concentration/contribution of the respective basis spectrum in the experimental data. To remove high frequency noise prior to spectral decomposition the experimental data was filtered by approximating it by a 40 degree polynomial, closer to the actual information content in the data[42]. The concentration of the species were assumed to follow sequential mono exponential kinetics. The data could be well represented by two components, one with a rise time $\tau_1 \approx 2$ µs and one with the rise time $\tau_2 \approx 250$ ms. The time dependent concentration of the two species is defined by Eqs. (2a) and (2b).

$$C_1 = \frac{\tau_2}{\tau_1 - \tau_2}\left(e^{-t/\tau_1} - e^{-t/\tau_2}\right) \tag{2a}$$

$$C_2 = 1 + \frac{\tau_2 e^{-t/\tau_2} - \tau_1 e^{-t/\tau_1}}{\tau_1 - \tau_2} \tag{2b}$$

To perform structural analysis of the difference scattering data a range of trial structures were generated using MD simulations (see subsequent section). For each of these structures the theoretical X-ray scattering was calculated using *CRYSOL*[43]. The scattering was calculated in 101 points, in the range 0 Å⁻¹ < q < 1 Å⁻¹, explicitly accounting for the hydrogen atoms present in the structure. The theoretical scattering was convoluted with the undulator spectrum, to allow comparison between experimental and theoretic scattering. The structures were considered to belong to a pool of dark or light candidate structures, depending on the state of FMN and cysteine 62 and the conformation of the LOV domain in the simulation. Every possible light-dark combination was calculated and scored against the experimental data using a previously used R factor[24], defined by Eq. (3).

$$R = \frac{\sum_q \sqrt{\left(\Delta I_{\mathrm{experiment}} - c\Delta I_{\mathrm{theory}}\right)^2}}{\sum_q \sqrt{\Delta I_{\mathrm{experiment}}^2}} \tag{3}$$

where the scaling parameter $c$ was allowed to vary. To avoid over fitting by allowing $c$ to scale completely free the experimental scattering was first approximately scaled to the theoretical scattering by comparing absolute (rather than difference) scattering curves. Based on the laser energy titration performed it was also known that the scattering curve represented ca. 70–75% turnover (Supplementary Fig. 4). The scaling parameter $c$ was therefore allowed to vary so that $0.35 \leq c \leq 1.6$. To avoid a disproportional influence on the fit by flexible regions, the histidine residues in the His-tag and residues 1–10 were removed after the simulation, but before calculating the scattering. After the difference scattering of all possible trial pairs were compared to the experimental data the 100 pairs with lowest R pairs were accepted as good fits.

**Generation of trial structures.** To generate physically reasonable trial structures molecular dynamics simulations were utilized. The simulations were performed using GROMACS 5.0.4[44] and the Charmm27 force field[45]. Force field parameters for FMN and the flavin-cysteinyl adduct were manually adapted from[46] to the GROMACS format. As a starting structure the YF1 crystal structure (pdb ID: 4GCZ) was used. Missing residues were added using MODELLER[47]. The ADP

found in the crystal structure was removed. There were no Adenosine nucleotides present during the simulations. The protein was placed in a cubic box, 1 nm larger than the protein in all directions and solvated with TIP3P water. The system was neutralized by adding 38 sodium ions. After adding ions the system underwent initial energy minimization until all forces were below 1000 kJ mol$^{-1}$ nm$^{-1}$. Subsequently the system was minimized for 100 ps in the NVT and NPT ensembles. During equilibration all non hydrogen atoms were position restrained with force constants of 1000 kJ mol$^{-1}$ nm$^{-2}$.

The final production runs included restraining the alpha carbons of the sensory domain to the positions where they were found according to ref. [22]. This was achieved by aligning alpha carbons of the best dark and light LOV domain structures from[22] to the LOV domain of YF1. The position of the alpha carbons was fixated using force constants of 10,000 kJ mol$^{-1}$ nm$^{-2}$. This generates two simulation trajectories, one where the LOV domain is locked in the dark state, and one where the LOV domain is locked in the light state. The simulations were run for ca. 450 ns. The atomic coordinates were saved every 10 ps to generate structures for the structural fitting routine. All the simulations were run with periodic boundary conditions in all directions, all bonds were constrained using the LINCS algorithm and a time step of 2 fs was used. Particle Mesh Ewald (PME) electrostatics with fourth order interpolation and with a grid spacing of 0.16 were used. The cutoff scheme was Verlet with a 1.0 nm cutoff, cutoffs for short-range electrostatic and van der Waals interactions were 1.0 nm as well. During the production run pressure control was achieved using the Parrinello-Rahman barostat ($\tau_P = 2$ ps, $P = 1$ bar) and temperature control was achieved via the modified Berendsen (velocity-rescale) thermostat ($\tau_T = 0.1$ ps, $T = 300$ K).

**Data availability**. All relevant data are available from figshare (doi: 10.6084/m9.figshare.5047990).

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

## Acknowledgements

This research used resources of the Advanced Photon Source, a U.S. Department of Energy (DOE) Office of Science User Facility operated for the DOE Office of Science by Argonne National Laboratory under Contract No. DE-AC02-06CH11357. Use of BioCARS was also supported by the National Institute of General Medical Sciences of the National Institutes of Health under grant number R24GM111072. The time-resolved X-ray diffraction set-up at APS Sector 14 was funded in part through a collaboration with Philip Anfinrud (NIH/NIDDK). We acknowledge the Paul Scherrer Institut, Villigen, Switzerland for provision of synchrotron radiation beamtime at beamline X12SA/cSAXS of the SLS. We acknowledge the European Synchrotron Radiation Facility and the beamline ID09B for provision of beamtime. The simulations were performed on resources provided by the Swedish National Infrastructure for Computing (SNIC) at NSC. S.W. acknowledges funding from the Swedish Foundation for International Cooperation in Research and Higher Education, the European Resereach Council, and the Foundation of Strategic Research, Sweden. A.M. acknowledges support from Deutsche Forschungsgemeinschaft through DFG grant MO2192/3-1 and a Sofja-Kovalevskaya Award by Alexander-von-Humboldt Foundation. H.T. acknowledges Academy of Finland (grant 285461) and the Finnish Cultural Foundation (grant 0131067). S.W. and M.H. acknowledges funding by the European Commission through FP7-PEOPLE-2013-IEF 624864.

## Author contributions

O.B., R.P.D., A.M., and S.W. designed the research and planned the experiments. R.P.D. expressed the proteins. R.P.D., S.K., and J.H. performed the absorption spectroscopy experiments and analyzed the data. O.B., L.H., M.R.P., A.H., A.B., H.T., E.G., M.H., J.A.I., S.N., A.Me., I.K., R.H., M.L., and S.W. performed the X-ray solution scattering experiments and O.B. analyzed the data and performed the structural modeling. O.B., A.M., and S.W. wrote the paper with input from all other authors.
