## [Peer Review file · Nature Communications]

Reviewers' comments:

Reviewer #1 (Remarks to the Author):

The manuscript by Berntsson et al. presents a time-resolved X-ray solution scattering investigation of the light-induced conformational changes occurring in the model sensor histidine kinase (SHK) YF1. The X-ray scattering experiments are complemented by time-resolved optical absorption spectroscopy measurements and are interpreted with the aid of calculations based on molecular dynamics simulations performed on both the light-activated and resting state of YF1. On the basis of the reported experimental data and of the performed structural analysis, the authors conclude that the coiled coil linker and the attached kinase domains rotate within a few microseconds simultaneously with the formation of the flavin-cysteine thioadduct in the LOV domain. Moreover, the authors observe a change in the X-ray scattering signal developing with a time constant of 50 ms that they assign to an internal rearrangement of the kinase domains.

The topic of the paper is quite interesting and the idea of using time-resolved X-ray scattering to unveil the sequence of structural rearrangements of a SHK in solution seems appropriate and potentially able to yield important insights to the current debate on the mechanism by which SHKs modulate their activity. Nevertheless, the conclusions drawn by the authors are not sufficiently supported by the experimental data presented in the present version of the manuscript. Therefore, I cannot recommend the publication of the manuscript in Nature Communications unless the issues detailed below are properly addressed by the authors.

Major points

1) The two basis spectra (Fig. 1b) that the authors extracted from the analysis of their time-resolved X-ray scattering data in terms of a sequential kinetic model are rather similar. It is crucial to prove that indeed the differences between the two curves are significant. This is particularly important in view of the fact that data from nanoseconds to milliseconds have been collected with one instrument, while those from milliseconds to seconds with a different one. Even if the authors have corrected their data in order to take into account the different conditions at which the data have been collected (different X-ray energy, polychromatic vs. monochromatic mode, different laser excitation, ...), all the performed corrections are of course affected by errors. It is thus well possible that the signal changes (attributed to the formation of the YF1-fin state) developing in the millisecond time scale are affected by such errors.

In order to assess the significance of the observed signal changes, I recommend the following:

1a) Add the ordinate scale in Fig. 1a, Fig. 1b, Fig. S2, and Fig. S3. X-ray scattered intensities are typically reported in arbitrary units since they depend on the specific experimental conditions employed. Nevertheless, if the data are normalized at one of the isosbestic points of the water scattering signal (as the authors did), then the intensity difference values are expressed in relative units with respect to the water scattering intensity. Such values can thus be used to evaluate at least the order of magnitude of the observed changes.

1b) In order to prove that the two datasets can be merged accurately, the authors should show that X-ray scattering differences measured at the same time delay at the two different instruments are indeed identical (differences should be much smaller than those observed in the millisecond evolution of the data).

1c) The authors should explain the meaning of the labels in Fig. 1a. Does the "500 ms-1 s" label indicate that data collected between 500 ms and 1 s have been averaged together to produce the corresponding curve reported in the plot?

1d) What is the expected fraction of proteins in the illuminated sample volume undergoing the observed conformational change? On the basis of their data (X-ray scattering, optical

spectroscopy, laser energy titration) the authors should be able to roughly estimate this quantity.

2) Dependence on protein concentration. While it is true that X-ray scattering signals at wide-angles are typically less sensitive to differences in protein concentrations than those at low angles, even a small effect could be relevant for the interpretation of the YF1 data in view of the comments reported above.

On page 3, line 75, the authors state that "... any concentration dependence of the scattering curves was below the q-range used in the analysis (Fig. S3 ...)". However, Fig. S3 only shows that there is a big effect at small q-values (below 0.1 1/\AA). In order to prove their statement, the authors should show the data also in a $q \cdot \Delta I$ vs. q plot (as in Fig. 1) rather than in a ΔI vs. q one, and in a similar scale to that of Fig. 1.

Data in Fig. S3 are relative to concentrations up to 20 mg/ml, while the data reported in the main manuscript have been collected at 25 mg/ml. I am assuming that the data in Fig. S3 are differences between the scattering signal of the sample at a few seconds from photoexcitation minus the signal of the protein in the dark state (this is not explained neither in the manuscript nor in the supporting information). If analogous data at 25 mg/ml are available, they should also be reported in Fig. S3.

Minor points:

- On page 3, line 74, the authors state that "The data were not significantly affected by solvent heating (Fig. S2)" and in Fig. S2 caption it is written that "There is essentially no contribution of the heat signal in the blue light excited difference scattering curve". This initially lead to me to conclude that indeed no detectable contribution from the solvent heating was observed. However, in the SI (pag. 19, line 337) it is clearly stated that the solvent heating contribution was actually subtracted from the blue-light excited difference data ("Difference scattering curves representing pure heating of the protein solution were collected by exposing the protein solution to an infrared laser pulse ... These curves were then scaled to and subtracted from the blue light excited difference scattering curves.").

The authors should clearly state in the manuscript that the solvent heating contribution has been removed and that Fig. S2 actually shows that the removal procedure is effective.

- The Methods section is rather short (well below the journal requirements). I would suggest to move or add relevant information like the sample composition, heat signal subtraction procedure, wavelength, energy density and laser spot size used for the time-resolved X-ray scattering measurements, capillary diameter, flowing speed of the sample (...) from the SI to the Methods section.

- Page 3, line 60: "excited triple state" should be "excited triplet state"

- Page 7, line 123: "To further quantify the structural change further ..."

Reviewer #2 (Remarks to the Author):

The work entitled "Sequential α -helical supercoiling activates a sensor histidine kinase" presents a time resolved X-ray scattering, and Molecular Dynamic study of the structural changes that occur during the activation of histidine kinase YF1. The data main conclusion is that the coiled coil linker and the attached histidine kinase domains rotate within microseconds, and subsequently in a much slower second step, the kinase domains rearrange internally. The authors suggest that this motion could represents a general template for activating motions in Histidine Kinases. The work is technically well done, by a reference group in the field of Histidine Kinases, and attacks one of the most important but still open issues which concerns how the activation signal is transmitted through the protein. The results are impressive. However, there are several concerns regarding the presentation of the work to fully appreciate its potential, that need to be addressed before it is

suitable for publication in Nature Communications. Specific points are noted below:

1) The main issue with the present manuscript is that it is presented for review with two other submitted manuscripts as additional information, both of them by the same authors. The first one corresponds to an ELDOR spectroscopy and site directed spin labelling study of the LOV-YF1 showing that light induces rotation and splaying apart of the two LOV photosensors in the dimeric protein. The second is time resolved X-ray study of the LOV domain of YtvA (another His kinase similar to YF1) which allows to characterize the light and dark structural states of this domain, which shows the same splaying apart and relative rotation of the two monomers reported in the first. This information of how the LOV sensing domain changes between light and dark state is essential for understanding the present work, and thus, in my opinion, some of the data concerning the time resolved x-ray study of the LOV domain (Berntsson et. al.), should be added to the present manuscript. Perhaps even joining both works, since they use the same methodology.

2) The second important problem with the manuscript is that the description of the reference structures used for the MD simulation and thus to analyze the reported changes is not completely clear. It is key that the authors describe how the "dark" and "light" (page 6) structures were built, what was modelled, using which crystals as templates. What about the alignment, etc. Also important, how do these structures relate to kinase state (active, inactive etc.)

3) Also related to the previous point, in page 8 authors state that "We used the crystal structure of 144 YF1, which is kinked and has CA domains, which are asymmetrically arranged" Is this a kinase active or inactive structure?.

4) In methods authors state that "As a starting structure the YF1 crystal structure (pdb id: 4GCZ) was used." Does this structure correspond to full YF1 (i.e with LOV and HK domains) structure? Which kinase (on/off) and LOV (dark/light) state does it correspond to.

5) In page 23 the authors state that "The final production runs included restraining the alpha carbons of the sensory domain to the positions where they were found according to (Berntsson et al. Submitted)". This information is essential for understanding how the MD were done and what they represent, since they are the heart of the time resolved X-ray crystallography interpretation. Therefore authors should add it in the main text. Moreover, authors should explicitly explain why and to which conformations restraints were applied.

6) In the introduction the author introduce the dark adapted YF1 state, which is their reference during the whole manuscript. It is not clear and not mentioned at any point how the dark adapted state is related to the kinase activity of YF1. The same is true for light produced states (YF1int, YF1fin). In other words, which state of the LOV domain (dark, excited, etc.) corresponds to autokinase inactive, kinase competent (active?) and/or fosfotransfersase competent state.

7) Given the relevance of the MD simulations for the present works, I expected a more detailed analysis of them. I suggest authors to perform not only basic RMSD and RMSF analysis, but also an essential dynamics analysis that could allow to characterize the motions of each state. For example how do the structures presented in Figure 3 (c,d) and Figure 4 to the transition mode obtained from covariance matrix of the joint MDs?

8) In the discussion, authors should analyze and compare their results in the context of few active structures available for His kinases. In particular those coming from kinases displaying LOV domains as sensors (see for example recent works of crystal structures reported by Goldbaum and Alzaris group's).

9) There is no mention in the discussion of how the changes observed in the catalytic domain are

related to the position of the phosphorylatable Histidine in relation to the ATP gamma. A discussion on this issue is essential to understand how the YF1_{int} and YF1_{fin} relate to autokinase competent structure.

10) In Figure S1, authors should mark the presence of phosphorylatable His. Do the catalytic domains have ATP bound?

Reviewer #3 (Remarks to the Author):

This paper presents a real time investigation of changes in the macroscopic structure of the sensor histidine kinase model YF1, correlating x-ray scattering, optical spectroscopy and molecular dynamics calculations. It reaches the conclusion that a two state structure change occurs – the first contemporary with the well-known photochemical formation of the flavin-cysteine adduct and the second a slower reorganisation. On the basis of MD modelling of the x-ray data the results are interpreted as an initial supercoiling in the coil region followed by an unbinding rebinding step in the CA domain.

This is a well written paper on an important topic providing new insight into the mechanism of operation of SHKs. The conclusion are of wide and general importance. As such it deserves publication in Nature Comm. However, there are some matters to be addressed in a revised version.

The connection between figs 3 and 4 and the conclusion of a two-step mechanism with only sub 10 microsecond and multi millisecond time constants requires further discussion. Figure 4 is logical and what one might expect – a fast small scale structure change on optical pumping followed by a much slower larger scale reorganisation. However, it is not clear how the first step in Fig 4 matches the data. If I am interpreting Figure 3b correctly, the main structure changes already seems to occur in the sub 10 microsecond step (and are mainly localised on CA and DHp). There is no obvious change in the Jalpha region. If Fig 3 is the experimental result then this apparent large scale ultrafast structure change has to be explained.

Some of these problems may arise from an oversimplified kinetic model – the signal-to-noise is not good enough to exclude more complex schemes. Although no spectroscopic evidence was found in this paper for intermediate timescale structure changes, there is recent evidence from spectroscopy and dynamics that structure changes occur in other LOV domains over multiple time scales (Biochem 55, 3107; JPC Lett 7, 5472; JPC B 121, 1010). Could these explain the unexpected occurrence of an apparently large structure change in a microsecond i.e. imposition of a two-state mechanism causes some slower structure changes to appear in the 'fast' kinetics? On Page 7 paragraph on line 115 it is unclear (to a non-expert) which atoms are meant by Calpha? These were also the constrained atoms initially, so what is the role of that constraint - could another choice change the outcome? If so please justify the choice made?

Other matters

Figure S1 was really helpful in understanding the text (especially in relation to the introduction) and therefore could be included in the main paper.

At the top of p4 it will be helpful to state exactly what x-ray scattering measures.

Methods l 194 flowed, not flown (and somewhere the capillary diameter should be stated).

Reviewer #4 (Remarks to the Author):

With a combination of X-ray scattering and MD simulations, the authors of this manuscript characterize the structural changes involved in photoreception by the engineered YF1 LOV-type sensor histidine kinase. Complementing work in two related manuscripts provided, here we see that illumination triggers the formation of, two sequentially-populated states, YF1(int) and YF1(fin), correlating to photochemical LOV-flavin adduct formation and subsequent larger rearrangements in the protein. Using MD trajectories to generate structures consistent with the

scattering data, rearrangements on the order of 10-15Å are suggested as being involved in activation of this system.

This manuscript has several strengths, particularly in that it provides some useful, domain-level information on full-length protein that complement the two accompanying manuscripts and work in the field. The scattering data themselves (Fig 1) are also excellent, providing a very solid starting point for this work.

Taken together, these strengths lead this reviewer to support publication provided that several straightforward issues can be addressed in revision:

(moderate/high) – lines 110-114, 123-129: it would be helpful for the authors to comment further about what degree of “spatial resolution” would be appropriate to view their results with. The authors are nicely candid about the moderate fit of their models to SAXS data at higher q values, and qualitatively describe the impact of this on the resolution of the conformational changes (lines 110-114). However, this discussion is followed by discussions of domain reorientation with 0.1 degree precision (lines 123-129), which seems overly precise in light of the moderate fits at high q . Please comment.

(moderate) – Supporting Information and accompanying spots in the text: I could not find any descriptions of the nucleotide state of samples used for data collection (and MD calculations, if relevant), please provide

(minor) – Fig 3 / Fig S1 and lines 164-168 in text (and elsewhere): it would be worth showing the locations of ATP and phosphoacceptor histidine to help orient non-expert readers to how these locations are related to proposed movements

(minor) – lines 164-175: I think this discussion would benefit from citing two recent publications when discussing aspects of the signaling model here, particularly regarding the asymmetric activation of one of the two CA domains (nicely reviewed in MP Bhate et al., *Structure* 23(2015): 981) and in the importance of breaking DHP-CA interactions to activate kinase function (again reviewed in Bhate et al. 2015; I’d also suggest that my group’s own work on a monomeric LOV-HK (G Rivera-Cancel et al., *PNAS* 111(2014): 17839) suggests an even further generality of model indicated here).

(minor) – lines 158-163: From my standpoint, a strength of a signaling model is shown by its ability to indicate how mutations affect systems. The authors detail here how some insertions in the coiled coil can affect YF1 function, which is a sensible and clear implication of the signaling model here. Can the authors similarly comment on the arguably tougher case how point mutations – like the H22P YF1 mutant in the accompanying Engelhard et al manuscript – lead to constitutive activation?

Kevin H. Gardner

Reviewer #1 (Remarks to the Author):

The manuscript by Berntsson et al. presents a time-resolved X-ray solution scattering investigation of the light-induced conformational changes occurring in the model sensor histidine kinase (SHK) YF1. The X-ray scattering experiments are complemented by time-resolved optical absorption spectroscopy measurements and are interpreted with the aid of calculations based on molecular dynamics simulations performed on both the light-activated and resting state of YF1. On the basis of the reported experimental data and of the performed structural analysis, the authors conclude that the coiled coil linker and the attached kinase domains rotate within a few microseconds simultaneously with the formation of the flavin-cysteine thioadduct in the LOV domain. Moreover, the authors observe a change in the X-ray scattering signal developing with a time constant of 50 ms that they assign to an internal rearrangement of the kinase domains.

The topic of the paper is quite interesting and the idea of using time-resolved X-ray scattering to unveil the sequence of structural rearrangements of a SHK in solution seems appropriate and potentially able to yield important insights to the current debate on the mechanism by which SHKs modulate their activity. Nevertheless, the conclusions drawn by the authors are not sufficiently supported by the experimental data presented in the present version of the manuscript. Therefore, I cannot recommend the publication of the manuscript in Nature Communications unless the issues detailed below are properly addressed by the authors.

Response

We thank the referee for her/his positive view on the potential impact of the paper. We also thank her/him for the technical comments, which we address in detail below. By addressing them, the paper is placed on much sturdier footing.

Comment 1

1) The two basis spectra (Fig. 1b) that the authors extracted from the analysis of their time-resolved X-ray scattering data in terms of a sequential kinetic model are rather similar. It is crucial to prove that indeed the differences between the two curves are significant. This is particularly important in view of the fact that data from nanoseconds to milliseconds have been collected with one instrument, while those from milliseconds to seconds with a different one. Even if the authors have corrected their data in order to take into account the different conditions at which the data have been collected (different X-ray energy, polychromatic vs. monochromatic mode, different laser excitation, ...), all the performed corrections are of course affected by errors. It is thus well possible that the signal changes (attributed to the formation of the YF1-fin state) developing in the millisecond time scale are affected by such errors.

In order to assess the significance of the observed signal changes, I recommend the following:

1a) Add the ordinate scale in Fig. 1a, Fig. 1b, Fig. S2, and Fig. S3. X-ray scattered intensities are typically reported in arbitrary units since they depend on the specific experimental conditions employed. Nevertheless, if the data are normalized at one of the isosbestic points of the water scattering signal (as the authors did), then the intensity difference values are expressed in relative units with respect to the water scattering

intensity. Such values can thus be used to evaluate at least the order of magnitude of the observed changes.

1b) In order to prove that the two datasets can be merged accurately, the authors should show that X-ray scattering differences measured at the same time delay at the two different instruments are indeed identical (differences should be much smaller than those observed in the millisecond evolution of the data).

1c) The authors should explain the meaning of the labels in Fig. 1a. Does the “500 ms-1 s” label indicate that data collected between 500 ms and 1 s have been averaged together to produce the corresponding curve reported in the plot?

1d) What is the expected fraction of proteins in the illuminated sample volume undergoing the observed conformational change? On the basis of their data (X-ray scattering, optical spectroscopy, laser energy titration) the authors should be able to roughly estimate this quantity.

Response/changes

1a) As per the reviewer's suggestion, values have been added to the axes of Fig 1a, 1b, S2 and S3. (Note that the numbering of figures changed due to moving S1 into the main manuscript.)

1b) We agree with the reviewer, this is a valid point. Since the two instruments unfortunately have very poor temporal overlap we can't easily measure the same time delay at both instruments for YF1. However, we have added an additional figure no. S5 showing how similar the results are for the two instruments when measuring the isolated LOV photosensor domain of YF1 (YtvA-LOV). This sample does not show any evolution of the signal on millisecond time scales and therefore the data recorded at the APS (Biocars) on microsecond time scales and at cSAXS on millisecond time scales can be directly compared. Fig S5 shows that the peak positions are fully reproducible between the two beamlines and that the amplitudes are accurate to <5%, better at low q. The agreement between the curves is much higher than the differences between the two basis spectra.

1c) This information is now added to the figure legend.

1d) The excited fraction was approximately 70%, which we had evaluated using the laser power titration (Fig S4). This has now been clarified in the text.

Comment 2

2) Dependence on protein concentration. While it is true that X-ray scattering signals at wide-angles are typically less sensitive to differences in protein concentrations than those at low angles, even a small effect could be relevant for the interpretation of the YF1 data in view of the comments reported above.

On page 3, line 75, the authors state that “... any concentration dependence of the scattering curves was below the q-range used in the analysis (Fig. S3 ...)”. However, Fig. S3 only shows that there is a big effect at small q-values (below 0.1 1/Å). In order to prove their statement, the authors should show the data also in a $q \cdot \Delta I$ vs. q plot (as in Fig. 1) rather than in a ΔI vs. q one, and in a similar scale to that of Fig. 1.

Data in Fig. S3 are relative to concentrations up to 20 mg/ml, while the data reported in the main manuscript have been collected at 25 mg/ml. I am assuming that the data in Fig. S3 are differences between the scattering signal of the sample at a few seconds from photoexcitation minus the signal of the protein in the dark state (this is not explained

neither in the manuscript nor in the supporting information). If analogous data at 25 mg/ml are available, they should also be reported in Fig. S3.

Response/changes

Figure S3 was unclear and we thank the reviewer for noticing this. Figure S3 has now been changed according to the reviewer's suggestions, confirming that the concentration used in the experiments does not influence the scattering at $q > 0.07 \text{ \AA}^{-1}$. The protein concentrations that was used in the experiments was 20-23 mg/mL (in different runs, 25mg/mL was incorrect) and this is now explicitly stated in the materials and methods in the main paper.

Comment 3

- On page 3, line 74, the authors state that "The data were not significantly affected by solvent heating (Fig. S2)" and in Fig. S2 caption it is written that "There is essentially no contribution of the heat signal in the blue light excited difference scattering curve". This initially lead to me to conclude that indeed no detectable contribution from the solvent heating was observed. However, in the SI (pag. 19, line 337) it is clearly stated that the solvent heating contribution was actually subtracted from the blue-light excited difference data ("Difference scattering curves representing pure heating of the protein solution were collected by exposing the protein solution to an infrared laser pulse ... These curves were then scaled to and subtracted from the blue light excited difference scattering curves."). The authors should clearly state in the manuscript that the solvent heating contribution has been removed and that Fig. S2 actually shows that the removal procedure is effective.

Response/changes

We agree and we thank the reviewer for pointing out this unclarity. As the reviewer suspected at first, there is almost no heat contribution in the data. However, the pure-heat contribution was still recorded, scaled and subtracted as it is normally done for time-resolved WAXS measurements. Figure S2 shows the data before heat subtraction. This is now more clearly written in the text.

Comment 4

- The Methods section is rather short (well below the journal requirements). I would suggest to move or add relevant information like the sample composition, heat signal subtraction procedure, wavelength, energy density and laser spot size used for the time-resolved X-ray scattering measurements, capillary diameter, flowing speed of the sample (...) from the SI to the Methods section.

Response/changes

We agree; all text from the SI has now been incorporated in the main text Methods section.

Comment 5

- Page 3, line 60: "excited triple state" should be "excited triplet state"

Response/changes

This has now been adjusted. We have also added a more comprehensive introductory description of photochemical events.

Comment 6

- Page 7, line 123: "To further quantify the structural change further ..."

Response/changes

This has now been adjusted.

Reviewer #2 (Remarks to the Author):

The work entitled “Sequential α -helical supercoiling activates a sensor histidine kinase” presents a time resolved X-ray scattering, and Molecular Dynamic study of the structural changes that occur during the activation of histidine kinase YF1. The data main conclusion is that the coiled coil linker and the attached histidine kinase domains rotate within microseconds, and subsequently in a much slower second step, the kinase domains rearrange internally. The authors suggest that this motion could represents a general template for activating motions in Histidine Kinases. The work is technically well done, by a reference group in the field of Histidine Kinases, and attacks one of the most important but still open issues which concerns how the activation signal is transmitted through the protein. The results are impressive. However, there are several concerns regarding the presentation of the work to fully appreciate its potential, that need to be addressed before it is suitable for publication in Nature Communications. Specific points are noted below:

Response

We thank the reviewer #2 for her/his positive assessment of the potential impact of the paper. We have addressed her/his comments below and through revisions/additions in the text.

Comment 1

1) The main issue with the present manuscript is that it is presented for review with two other submitted manuscript as additional information, both of them by the same authors. The first one corresponds to an ELDOR spectroscopy and site directed spin labelling study of the LOV-YF1 showing that light induces rotation and splaying apart of the two LOV photosensors in the dimeric protein. The second is time resolved X-ray study of the LOV domain of YtvA (another His kinase similar to YF1) which allows to characterize the light and dark structural states of this domain, which shows the same splaying apart and relative rotation of the two monomers reported in the first. This information of how the LOV sensing domain changes between light and dark state is essential for understanding the present work, and thus, in my opinion, some of the data concerning the time resolved x-ray study of the LOV domain (Berntsson et. al.), should be added to the present manuscript. Perhaps even joining both works, since they use the same methodology.

Response/changes

We agree that the structural mode of activation of the LOV domains is important for understanding the current manuscript. The two manuscripts describing this use X-ray solution scattering and electron paramagnetic resonance spectroscopy, respectively. They are already in press at Structure and Scientific Reports, respectively (doi: 10.1016/j.str.2017.04.006 and 10.1038/s41598-017-01497-7).

The focus of these two manuscripts is on the events within the LOV photosensor, whereas the present manuscript concentrates on the downstream events of signal transduction to the histidine kinase effector and rearrangements thereof.

To facilitate the understanding of the present manuscript, we have added as Fig. 1b structural models of the isolated LOV photosensor dimer in dark-adapted and

light-adapted states. We have also added a description of these previous results to the end of the introduction section.
Comment 2
2) The second important problem with the manuscript is that the description of the reference structures used for the MD simulation and thus to analyze the reported changes is not completely clear. It is key that the authors describe how the “dark” and “light” (page 6) structures were built, what was modelled, using which crystals as templates. What about the alignment, etc. Also important, how do these structures relate to kinase state (active, inactive etc.)
Response/changes
We thank the reviewer for pointing this out. Indeed, information on how the structures and MD simulations were prepared was spread out over various places in the manuscript. This has now been revised and the procedure is more clearly described under “Structural Analysis” in the results section and in even more detail in the methods section.
Comment 3
3) Also related to the previous point, in page 8 author state that “We used the crystal structure of 144 YF1, which is kinked and has CA domains, which are asymmetrically arranged” Is this a kinase active or inactive structure?.
Response/changes
We appreciate this and the following comment and have added a more detailed description in the introduction. Biochemical data (Möglich et al. JMB 2009) show that in the dark YF1 is kinase-active and in the light it is phosphatase active. At the structural level, the kinase-active nature of the dark-adapted YF1 structure 4GCZ is arguably reflected in its kink and the asymmetry of the CA domains, features that are associated with the kinase-active state of sensor histidine kinases (cf. Bhate et al. Structure 2015). Notably, to date YF1 represents the only system for which both a full-length structure (i.e. sensor plus effector) and the molecular identity of the signal/stimulus are known.
Comment 4
4) In methods authors state that “As a starting structure the YF1 crystal structure (pdb id: 4GCZ) was used.” Does this structure corresponds to full YF1 (i.e with LOV and HK domains) structure? Which kinase (on/off) and LOV (dark/light) state does it correspond to.
Response/changes
Cf. previous comment.
Comment 5
5) In page 23 the authors state that “The final production runs included restraining the alpha carbons of the sensory domain to the positions where they were found according to (Berntsson et al. Submitted)”. This is information is essential for understanding how the MD were done and what they represent, since they are the heart of the time resolved

X-ray crystallography interpretation. Therefore authors should add it in the main text. Moreover, author should explicitly explain why and to which conformations restraints were applied.

Response/changes

This was not clearly written in the text and we thank the reviewer for pointing this out. We have now described the position restraints used in the MD simulations in more detail.

Comment 6

6) In the introduction the author introduce the dark adapted YF1 state, which is their reference during the whole manuscript. It is not clear and not mentioned at any point how the dark adapted state is related to the kinase activity of YF1. The same is true for light produced states (YF1_{int}, YF1_{fin}). In other words, which state of the LOV domain (dark, excited, etc.) corresponds to autokinase inactive, kinase competent (active?) and/or fosfotransfersase competent state.

Response/changes

According to biochemical data (cf. comment 4), YF1_{dark} is in its kinase-active state and the long-lived state produced by light absorption, YF1_{fin}, is in its phosphatase-state. We cannot assign a functional state to the short-lived intermediate YF1_{int}. A description of these assignments has been added to the Discussion section.

Comment 7

7) Given the relevance of the MD simualtions for the present works, I expected a more detailed analysis of them. I suggest authors to perform not only basic RMSD and RMSF analysis, but also an essential dynamics analysis that could allow to characterize the motions of each state. For example how do the structures presented in Figure 3 (c,d) and Figure 4 to the transition mode obtained from covariance matrix of the joint MDs?

Response/changes

We respectfully disagree that such as comparison is pertinent. In particular, we have not attempted to perform a stringent MD study, but rather used MD as a tool to generate a pool of physically reasonable structures, from which good fits to the experimental data can be selected.

We have attempted to do a principal component analysis of the good fits, but this failed to reveal any clear structural changes, since the variations in the resting state structures was too high. In a way, this effect is described in the paper, where it is stated that

“All pairs of structures were overlaid and visually inspected to identify a common structural change, but the resting and photoproduct state structures did not group into two distinguishable sets of structures. Considering that the data report on differences rather than on absolute structures this is not very surprising.”

We would prefer to keep the existing text, since addition of an extra level of analysis to show something which is already seen by visually inspecting the structures, would not make the paper easier to understand.

Comment 8

8) In the discussion, authors should analyze and compare their results in the context of few active structures available for His kinases. In particular those coming from kinases displaying LOV domains as sensors (see for example recent works of crystal structures reported by Goldbaum and Alzaris group's).

Response/changes

We agree that our results were not properly put in the scientific context, and we have thus revised our manuscript in the discussion section to refer to the indicated papers plus other pertinent papers. Also cf. comments by reviewer #4.

Comment 9

9) There is no mention in the discussion of how the changes observed in the catalytic domain are related to the position of the phosphorylatable Histidine in relation to the ATP gamma. A discussion for this issue is essential to understand how the YF1int and YF1fin relate to autokinase competent structure.

Response/changes

We agree that our initial description of these aspects was lacking and we have hence revised the discussion section. The arguably clearest depiction of the likely events manifesting in the YF1 DHp/CA domains after light absorption is provided by Fig. 4C of Trajtenberg et al. (2016) eLife. In a crystal structure the phosphatase-active state of DesK from *B. subtilis* could be trapped and structurally resolved. This state differs from the kinase-active state in the rotation of the DHp helices, exactly as we presently invoke for YF1. As a corollary, the active-site histidine is rotated and sequestered in the interior of the DHp antiparallel four-helix bundle. Concomitantly, the residues required for phosphatase activity are moved into place. In response to this comment, we have largely restructured and significantly expanded the entire Discussion section.

Comment 10

10) In Figure S1, authors should mark the presence of phosphorylatable His. Do the catalytic domains have ATP bound?

Response/changes

The phosphorylatable histidine is now marked in the figure, which has moved to become Figure 1. The position that the adenosine nucleotides would occupy is marked, but these are not present in the experiment or the simulations. This is now described more clearly in the manuscript.

Reviewer #3 (Remarks to the Author):

This paper presents a real time investigation of changes in the macroscopic structure of the sensor histidine kinase model YF1, correlating x-ray scattering, optical spectroscopy and molecular dynamics calculations. It reaches the conclusion that a two state structure change occurs – the first contemporary with the well-known photochemical formation of the flavin-cysteine adduct and the second a slower reorganisation. On the basis of MD modelling of the x-ray data the results are interpreted as an initial supercoiling in the coil region followed by an unbinding rebinding step in the CA domain.

This is a well written paper on an important topic providing new insight into the mechanism of operation of SHKs. The conclusion are of wide and general importance. As such it deserves publication in Nature Comm. However, there are some matters to be addressed in a revised version.

Response

We appreciate the encouraging review and have revised our manuscript according to the suggestions.

Comment 1

The connection between figs 3 and 4 and the conclusion of a two-step mechanism with only sub 10 microsecond and multi millisecond time constants requires further discussion. Figure 4 is logical and what one might expect – a fast small scale structure change on optical pumping followed by a much slower larger scale reorganisation. However, it is not clear how the first step in Fig 4 matches the data. If I am interpreting Figure 3b correctly, the main structure changes already seems to occur in the sub 10 microsecond step (and are mainly localised on CA and DHp). There is no obvious change in the Jalpha region. If Fig 3 is the experimental result then this apparent large scale ultrafast structure change has to be explained.

Some of these problems may arise from an oversimplified kinetic model – the signal-to-noise is not good enough to exclude more complex schemes. Although no spectroscopic evidence was found in this paper for intermediate timescale structure changes, there is recent evidence from spectroscopy and dynamics that structure changes occur in other LOV domains over multiple time scales (Biochem 55, 3107; JPC Lett 7, 5472; JPC B 121, 1010). Could these explain the unexpected occurrence of an apparently large structure change in a microsecond i.e. imposition of a two-state mechanism causes some slower structure changes to appear in the ‘fast’ kinetics?

Response/changes

We concur that the kinetic model that the X-ray scattering data yield should be considered a minimal model. As always in kinetics, more complex models with additional intermediates can be envisioned. Different experimental techniques may be more or less able to pick up such additional intermediates. We change the text by mentioning these issues and by citing pertinent papers that observed more complex schemes for photoreceptors when studying them by other experimental techniques.

The reanalysis of the data has yielded slightly different basis spectra for the two states, but the structural interpretation (left-handed supercoiling in both states and an additional rearrangement of the HK module in YF1_fin) are unaffected. A 2 microsecond

rearrangement of the entire protein as observed by us is in our opinions not unusually fast - it is fully within expectation for similar structural changes. It appears that changes in the LOV domain directly transduce through the entire YF1 protein. We have now state this much clearer in the results section.

Comment 2

On Page 7 paragraph on line 115 it is unclear (to a non-expert) which atoms are meant by Calpha? These were also the constrained atoms initially, so what is the role of that constraint - could another choice change the outcome? If so please justify the choice made?

Response/changes

The position restraints and their purpose were not clearly described in the text. We have now changed the text to make it more accessible. We thank the reviewer for pointing this out.

Comment 3

Figure S1 was really helpful in understanding the text (especially in relation to the introduction) and therefore could be included in the main paper.

Response/changes

We thank the reviewer for this suggestion which is implemented in the revised manuscript. In response to reviewer #2, we have also added to this figure a brief summary of the light-induced structural changes for the isolated YF1 LOV photosensor, as previously observed by X-ray solution scattering and electron paramagnetic resonance, cf. above response to reviewer #2.

Comment 4

At the top of p4 it will be helpful to state exactly what x-ray scattering measures.

Response/changes

Such an introduction has now been added. We thank the reviewer for suggesting this.

Comment 5

Methods I 194 flowed, not flown (and somewhere the capillary diameter should be stated).

Response/changes

This has been changed.

Reviewer #4 (Remarks to the Author):

With a combination of X-ray scattering and MD simulations, the authors of this manuscript characterize the structural changes involved in photoreception by the engineered YF1 LOV-type sensor histidine kinase. Complementing work in two related manuscripts provided, here we see that illumination triggers the formation of, two sequentially-populated states, YF1(int) and YF1(fin), correlating to photochemical LOV-flavin adduct formation and subsequent larger rearrangements in the protein. Using MD trajectories to generate structures consistent with the scattering data, rearrangements on the order of 10-15Å are suggested as being involved in activation of this system.

This manuscript has several strengths, particularly in that it provides some useful, domain-level information on full-length protein that complement the two accompanying manuscripts and work in the field. The scattering data themselves (Fig 1) are also excellent, providing a very solid starting point for this work.

Taken together, these strengths lead this reviewer to support publication provided that several straightforward issues can be addressed in revision:

Response

We are grateful for Kevin Gardner's endorsement and helpful comments all of which we implement in the revised manuscript.

Comment 1

(moderate/high) – lines 110-114, 123-129: it would be helpful for the authors to comment further about what degree of “spatial resolution” would be appropriate to view their results with. The authors are nicely candid about the moderate fit of their models to SAXS data at higher q values, and qualitatively describe the impact of this on the resolution of the conformational changes (lines 110-114). However, this discussion is followed by discussions of domain reorientation with 0.1 degree precision (lines 123-129), which seems overly precise in light of the moderate fits at high q. Please comment.

Response/changes

This is an interesting point. It is true that there is a roughly inverse relationship between q-scale and length scale. Changes at low q represent global rearrangements, even if these are very small. As an example sub-Å changes in the radius of gyration are characterized by strong difference signals at low q (eg. [Levantino, Nat. Commun. 2015]). The values of domain reorientation are based on the structural models that fit the scattering data and as such they have a very high precision. How these values are calculated has now been clarified in the text. Nevertheless, we agree that reporting values with 0.1 degrees precision gives a false sense of spatial resolution, this has now been changed to full degrees.

Comment 2

(moderate) – Supporting Information and accompanying spots in the text: I could not find any descriptions of the nucleotide state of samples used for data collection (and MD calculations, if relevant), please provide

Response/changes
We have now clarified that no adenine nucleotides were present in the experiments or simulations.
Comment 3
(minor) – Fig 3 / Fig S1 and lines 164-168 in text (and elsewhere): it would be worth showing the locations of ATP and phosphoacceptor histidine to help orient non-expert readers to how these locations are related to proposed movements
Response/changes
The location of the adenine nucleotides as well as the phosphoacceptor histidine has now been added to Fig S1, which is now Fig 1.
Comment 4
(minor) – lines 164-175: I think this discussion would benefit from citing two recent publications when discussing aspects of the signaling model here, particularly regarding the asymmetric activation of one of the two CA domains (nicely reviewed in MP Bhate et al., Structure 23(2015): 981) and in the importance of breaking DHp-CA interactions to activate kinase function (again reviewed in Bhate et al. 2015; I'd also suggest that my group's own work on a monomeric LOV-HK (G Rivera-Cancel et al., PNAS 111(2014): 17839) suggests an even further generality of model indicated here).
Response/changes
We again agree and have expanded the Discussion section accordingly. Also cf. our responses to comments by reviewer #3.
Comment 5
(minor) – lines 158-163: From my standpoint, a strength of a signaling model is shown by its ability to indicate how mutations affect systems. The authors detail here how some insertions in the coiled coil can affect YF1 function, which is a sensible and clear implication of the signaling model here. Can the authors similarly comment on the arguably tougher case how point mutations – like the H22P YF1 mutant in the accompanying Engelhard et al manuscript – lead to constitutive activation?
Response/changes
We expanded our discussion of functional YF1 variants and also accommodate H22P now. In a nutshell, this variant is highly interesting as a single residue exchange suffices for inverting signal polarity (cf. Diensthuber et al. 2013 Structure): that is, in H22P the kinase-active state is not assumed in dark (as is the case for YF1) but following blue-light exposure. Electron paramagnetic resonance (described in the just-published Engelhard work) showed that H22P has a drastically altered dimer interface yet undergoes similar light-induced pivot motions. In marked difference to YF1, these motions lead to an approach of the J α anchor sites, rather than to a moving apart. In the context of the present findings, this motion with opposite sign in H22P might trigger supercoiling with opposite sign than in YF1, i.e. right-handed supercoiling upon light exposure. We note that

while entirely plausible this mechanism for H22P should be considered speculative at present.

Reviewers' comments:

Reviewer #1 (Remarks to the Author):

The authors have satisfactorily addressed all the points that I have raised on the previous version of the manuscript.

My main concern was on the significance of the observed X-ray scattering signal changes. From the revised figures it is now possible to estimate that the amplitude of scattering difference signals is of the order of 1.5×10^{-3} with respect to the absolute scattering at 1.4-1.6 Å⁻¹. This is well within the sensitivity of current instrumentation for time-resolved X-ray scattering measurements. For example, in the case of the monomeric ~18 kDa protein myoglobin, Levantino et al. (Nat. Commun. 2015) reported a difference signal amplitude in the WAXS region of the order of 5×10^{-4} at a protein concentration of ~40 mg/ml.

On the other hand, the authors have now clarified that the experimental data reported in the revised Fig. 2a are averaged over several time-delays. In particular, one of the curve reported in Fig. 2a is the result of an average over data collected in the "5 μs-10 ms" time-range (i.e., more than 3 order of magnitude in time). In order to allow the reader to evaluate the signal changes within the noise the authors should add a further supporting figure with a comparison of two difference patterns at two relevant time-delays (e.g. 100 μs and 1 s) where the YF1_{int.} and YF1_{fin.} states are, respectively, fully populated.

Minor point: typo in Fig. 2a - the label "500-1.5 μs" should be "0.5-1.5 μs"

Before recommending publication in Nature Communications, I would like to see the above mentioned figure and a revised version of the paper including a comment on the significance of the observed X-ray signal changes.

Reviewer #2 (Remarks to the Author):

The revised version of "Sequential α-helical supercoiling activates a sensor histidine kinase" presents a significant improved manuscript. Specifically, in the revised version, in order to improve the understanding of the present manuscript in relation to how the LOV sensing domain changes between light and dark state, the authors added (in Fig 1b) structural models of the isolated LOV photosensor dimer in dark-adapted and light-adapted states, and a description of the results presented in -now published works- in the introduction. The authors also added key new segment "Structural analysis" (pages 7 to 10) where they described in adequate detail how the "dark" and "light" adapted structures used for MD were generated, and how the restraints were applied. They also clarified that the dark YF1 state is kinase active (represented by pdbid 4GCZ and showing a kink and assymmetric CA domains), and in the light state it displays phosphatase activity. Finally, they also commented on recent structural works on SHK activation mechanism from other groups. In summary, most of my concerns have been correctly addressed. I have only a few minor comments (noted below) and strongly believe the work is suitable for publication.

1) The authors may add a comment stating why they simulated, and thus interpreted, their data using for both, dark and light, states SHK domain structures corresponding to dark, autokinase competent, state. Why not use for the light states an inactive (or phosphatase competent) SHK structure?

2) In the introduction (page 3) authors could add information referring to the structure in the dark autokinase competente state, characterized by the kink and assymmetric CA domains.

Reviewer #3 (Remarks to the Author):

The reviewers raised a number of points mainly concerning the analysis and presentation of the results. For those where I am competent to judge those points have been carefully addressed by the authors. Certainly the additional text makes it easier to see what is claimed. It still seems surprising that the structure change in fig 6 (b) occurs simultaneously with thioadduct formation in the flavin but the data support this; the result will certainly stimulate further research. Thus I believe the paper warrants publication in Nat.Comm.

There are only some typos to address (lines 94 peaks not pekass, line 208 within microseconds of, not after)

Reviewer #1 (Remarks to the Author):

The authors have satisfactorily addressed all the points that I have raised on the previous version of the manuscript.

My main concern was on the significance of the observed X-ray scattering signal changes. From the revised figures it is now possible to estimate that the amplitude of scattering difference signals is of the order of 1.5×10^{-3} with respect to the absolute scattering at 1.4-1.6 Å⁻¹. This is well within the sensitivity of current instrumentation for time-resolved X-ray scattering measurements. For example, in the case of the monomeric ~18 kDa protein myoglobin, Levantino et al. (Nat. Commun. 2015) reported a difference signal amplitude in the WAXS region of the order of 5×10^{-4} at a protein concentration of ~40 mg/ml.

On the other hand, the authors have now clarified that the experimental data reported in the revised Fig. 2a are averaged over several time-delays. In particular, one of the curve reported in Fig. 2a is the result of an average over data collected in the “5 μs-10 ms” time-range (i.e., more than 3 order of magnitude in time). In order to allow the reader to evaluate the signal changes within the noise the authors should add a further supporting figure with a comparison of two difference patterns at two relevant time-delays (e.g. 100 μs and 1 s) where the YF1_{int.} and YF1_{fin.} states are, respectively, fully populated.

Minor point: typo in Fig. 2a - the label “500-1.5 μs” should be “0.5-1.5 μs”

Before recommending publication in Nature Communications, I would like to see the above mentioned figure and a revised version of the paper including a comment on the significance of the observed X-ray signal changes.

Response

We thank the reviewer for the positive comments on the manuscript. A figure showing the scattering at 50 microseconds as well as the average scattering for 3-4 s has now been added, along with a comment on the significance of the changes in the X-ray scattering signal. The new SI figure nicely illustrates the different patterns in the data that give rise to the two states in the model.

Reviewer #2 (Remarks to the Author):

The revised version of "Sequential α -helical supercoiling activates a sensor histidine kinase" presents a significant improved manuscript. Specifically, in the revised version, in order to improve the understanding of the present manuscript in relation to how the LOV sensing domain changes between light and dark state, the authors added (in Fig 1b) structural models of the isolated LOV photosensor dimer in dark-adapted and light-adapted states, and a description of the results presented in -now published works- in the introduction. The authors also added key new segment "Structural analysis" (pages 7 to 10) where they described in adequate detail how the "dark" and "light" adapted structures used for MD were generated, and how the restraints were applied. They also clarified that the dark YF1 state is kinase active (represented by pdbid 4GCZ and showing a kink and assymetric CA domains), and in the light state it displays phosphatase activity. Finally, they also commented on recent structural works on SHK activation mechanism from other groups. In summary, most of my concerns have been correctly addressed. I have only a few minor comments (noted below) and strongly believe the work is suitable for publication.

Response

We are glad that our revisions have been satisfactory. The two comments are addressed below.

Comment 1

1) The authors may add a comment stating why they simulated, and thus interpreted, their data using for both, dark and light, states SHK domain structures corresponding to dark, autokinase competent, state. Why not use for the light states an inactive (or phosphatase competent) SHK structure?

Response/changes

To the best of our knowledge, there are no structures available of SHKs comprising both sensor and kinase domain both in presence and absence of its signal. Any such model would thus have to be based on additional assumptions and modeling which would strongly bias the result.

Comment 2

2) In the introduction (page 3) authors could add information referring to the structure in the dark autokinase competente state, characterized by the kink and assymetric CA domains.

Response/changes

This information is now added to the introduction. We thank the reviewer for suggesting this as it makes it easier to follow the text.

Reviewer #3 (Remarks to the Author):

The reviewers raised a number of points mainly concerning the analysis and presentation of the results. For those where I am competent to judge those points have been carefully addressed by the authors. Certainly the additional text makes it easier to see what is claimed. It still seems surprising that the structure change in fig 6 (b) occurs simultaneously with thioadduct formation in the flavin but the data support this; the result will certainly stimulate further research. Thus I believe the paper warrants publication in Nat.Comm.

There are only some typos to address (lines 94 peaks not pekass, line 208 within microseconds of, not after)

Response

We thank the reviewer for this positive assessment and for pointing out the typos above which have now been corrected.

REVIEWERS' COMMENTS:

Reviewer #1 (Remarks to the Author):

The authors have satisfactorily addressed all my concerns. The paper is now ready for publication.